# A Block Coordinate Ascent Algorithm for Mean-Variance Optimization

**Tengyang Xie**[*]
UMass Amherst
txie@cs.umass.edu

**Bo Liu**[*]
Auburn University
boliu@auburn.edu

**Yangyang Xu**
Rensselaer Polytechnic Institute
xuy21@rpi.edu

**Mohammad Ghavamzadeh**
Facebook AI Research
mgh@fb.com

**Yinlam Chow**
Google DeepMind
yinlamchow@google.com

**Daoming Lyu**
Auburn University
daoming.lyu@auburn.edu

**Daesub Yoon**
ETRI
eyetracker@etri.re.kr

## Abstract

Risk management in dynamic decision problems is a primary concern in many fields, including financial investment, autonomous driving, and healthcare. The mean-variance function is one of the most widely used objective functions in risk management due to its simplicity and interpretability. Existing algorithms for mean-variance optimization are based on multi-time-scale stochastic approximation, whose learning rate schedules are often hard to tune, and have only asymptotic convergence proof. In this paper, we develop a model-free policy search framework for mean-variance optimization with finite-sample error bound analysis (to local optima). Our starting point is a reformulation of the original mean-variance function with its Legendre-Fenchel dual, from which we propose a stochastic block coordinate ascent policy search algorithm. Both the asymptotic convergence guarantee of the last iteration's solution and the convergence rate of the randomly picked solution are provided, and their applicability is demonstrated on several benchmark domains.

## 1 Introduction

Risk management plays a central role in sequential decision-making problems, common in fields such as portfolio management [Lai et al., 2011], autonomous driving [Maurer et al., 2016], and health-care [Parker, 2009]. A common risk-measure is the variance of the expected sum of rewards/costs and the mean-variance trade-off function [Sobel, 1982; Mannor and Tsitsiklis, 2011] is one of the most widely used objective functions in risk-sensitive decision-making. Other risk-sensitive objectives have also been studied, for example, Borkar [2002] studied exponential utility functions, Tamar et al. [2012] experimented with the Sharpe Ratio measurement, Chow et al. [2018] studied value at risk (VaR) and mean-VaR optimization, Chow and Ghavamzadeh [2014], Tamar et al. [2015b], and Chow et al. [2018] investigated conditional value at risk (CVaR) and mean-CVaR optimization in a static setting, and Tamar et al. [2015a] investigated coherent risk for both linear and nonlinear system dynamics. Compared with other widely used performance measurements, such as the Sharpe Ratio and CVaR, the mean-variance measurement has explicit interpretability and computational

---

[*]Equal contribution. Corresponding to: boliu@auburn.edu

advantages [Markowitz et al., 2000; Li and Ng, 2000]. For example, the Sharpe Ratio tends to lead to solutions with less mean return [Tamar et al., 2012]. Existing mean-variance reinforcement learning (RL) algorithms [Tamar et al., 2012; Prashanth and Ghavamzadeh, 2013, 2016] often suffer from heavy computational cost, slow convergence, and difficulties in tuning their learning rate schedules. Moreover, all their analyses are asymptotic and no rigorous finite-sample complexity analysis has been reported. Recently, Dalal et al. [2018] provided a general approach to compute finite sample analysis in the case of linear multiple time scales stochastic approximation problems. However, existing multiple time scales algorithms like [Tamar et al., 2012] consist of nonlinear term in its update, and cannot be analyzed via the method in Dalal et al. [2018]. All these make it difficult to use them in real-world problems. The goal of this paper is to propose a mean-variance optimization algorithm that is both computationally efficient and has finite-sample analysis guarantees. This paper makes the following contributions: **1)** We develop a computationally efficient RL algorithm for mean-variance optimization. By reformulating the mean-variance function with its Legendre-Fenchel dual [Boyd and Vandenberghe, 2004], we propose a new formulation for mean-variance optimization and use it to derive a computationally efficient algorithm that is based on stochastic cyclic block coordinate descent. **2)** We provide the sample complexity analysis of our proposed algorithm. This result is novel because although cyclic block coordinate descent algorithms usually have empirically better performance than randomized block coordinate descent algorithms, yet almost all the reported analysis of these algorithms are asymptotic [Xu and Yin, 2015].

Here is a roadmap for the rest of the paper. Section 2 offers a brief background on risk-sensitive RL and stochastic variance reduction. In Section 3, the problem is reformulated using the Legendre-Fenchel duality and a novel algorithm is proposed based on stochastic block coordinate descent. Section 4 contains the theoretical analysis of the paper that includes both asymptotic convergence and finite-sample error bound. The experimental results of Section 5 validate the effectiveness of the proposed algorithms.

## 2 Backgrounds

This section offers a brief overview of risk-sensitive RL, including the objective functions and algorithms. We then introduce block coordinate descent methods. Finally, we introduce the Legendre-Fenchel duality, the key ingredient in formulating our new algorithms.

### 2.1 Risk-Sensitive Reinforcement Learning

Reinforcement Learning (RL) [Sutton and Barto, 1998] is a class of learning problems in which an agent interacts with an unfamiliar, dynamic, and stochastic environment, where the agent's goal is to optimize some measures of its long-term performance. This interaction is conventionally modeled as a Markov decision process (MDP), defined as the tuple $(\mathcal{S}, \mathcal{A}, P_0, P_{ss'}^a, r, \gamma)$, where $\mathcal{S}$ and $\mathcal{A}$ are the sets of states and actions, $P_0$ is the initial state distribution, $P_{ss'}^a$ is the transition kernel that specifies the probability of transition from state $s \in \mathcal{S}$ to state $s' \in \mathcal{S}$ by taking action $a \in \mathcal{A}$, $r(s, a) : \mathcal{S} \times \mathcal{A} \to \mathbb{R}$ is the reward function bounded by $R_{\max}$, and $0 \leq \gamma < 1$ is a discount factor. A *parameterized stochastic policy* $\pi_\theta(a|s) : \mathcal{S} \times \mathcal{A} \to [0, 1]$ is a probabilistic mapping from states to actions, where $\theta$ is the tunable parameter and $\pi_\theta(a|s)$ is a differentiable function w.r.t. $\theta$.

One commonly used performance measure for policies in *episodic* MDPs is the *return* or cumulative sum of rewards from the starting state, i.e., $R = \sum_{k=1}^{\tau} r(s_k, a_k)$, where $s_1 \sim P_0$ and $\tau$ is the first passage time to the recurrent state $s^*$ [Puterman, 1994; Tamar et al., 2012], and thus, $\tau := \min\{k > 0 \mid s_k = s^*\}$. In risk-neutral MDPs, the algorithms aim at finding a near-optimal policy that maximizes the expected sum of rewards $J(\theta) := \mathbb{E}_{\pi_\theta}[R] = \mathbb{E}_{\pi_\theta}\left[\sum_{k=1}^{\tau} r(s_k, a_k)\right]$. We also define the square-return $M(\theta) := \mathbb{E}_{\pi_\theta}[R^2] = \mathbb{E}_{\pi_\theta}\left[\left(\sum_{k=1}^{\tau} r(s_k, a_k)\right)^2\right]$. In the following, we sometimes drop the subscript $\pi_\theta$ to simplify the notation.

In risk-sensitive mean-variance optimization MDPs, the objective is often to maximize $J(\theta)$ with a variance constraint, i.e.,

$$\max_\theta \quad J(\theta) = \mathbb{E}_{\pi_\theta}[R]$$
$$\text{s.t.} \quad \text{Var}_{\pi_\theta}(R) \leq \zeta, \tag{1}$$

where $\mathrm{Var}_{\pi_\theta}(R) = M(\theta) - J^2(\theta)$ measures the variance of the return random variable $R$, and $\zeta > 0$ is a given risk parameter [Tamar et al., 2012; Prashanth and Ghavamzadeh, 2013]. Using the Lagrangian relaxation procedure [Bertsekas, 1999], we can transform the optimization problem (1) to maximizing the following unconstrained objective function:

$$\begin{aligned} J_\lambda(\theta) :=& \mathbb{E}_{\pi_\theta}[R] - \lambda\big(\mathrm{Var}_{\pi_\theta}(R) - \zeta\big) \\ =& J(\theta) - \lambda\big(M(\theta) - J(\theta)^2 - \zeta\big). \end{aligned} \tag{2}$$

It is important to note that the mean-variance objective function is *NP-hard* in general [Mannor and Tsitsiklis, 2011]. The main reason for the hardness of this optimization problem is that although the variance satisfies a Bellman equation [Sobel, 1982], unfortunately, it lacks the monotonicity property of dynamic programming (DP), and thus, it is not clear how the related risk measures can be optimized by standard DP algorithms [Sobel, 1982].

The existing methods to maximize the objective function (2) are mostly based on stochastic approximation that often converge to an equilibrium point of an ordinary differential equation (ODE) [Borkar, 2008]. For example, Tamar et al. [2012] proposed a policy gradient algorithm, a two-time-scale stochastic approximation, to maximize (2) for a fixed value of $\lambda$ (they optimize over $\lambda$ by selecting its best value in a finite set), while the algorithm in Prashanth and Ghavamzadeh [2013] to maximize (2) is actor-critic and is a three-time-scale stochastic approximation algorithm (the third time-scale optimizes over $\lambda$). The stochastic compositional optimization method [Wang et al., 2017] also needs two-time-scale stepsize tuning for mean-variance optimization, and dual embeddings [Dai et al., 2017] assume the embedded problem can be solved exactly. These approaches suffer from certain drawbacks: **1)** Most of the analyses of ODE-based methods are asymptotic, with no sample complexity analysis. **2)** It is well-known that multi-time-scale approaches are sensitive to the choice of the stepsize schedules, which is a non-trivial burden in real-world problems. **3)** The ODE approach does not allow extra penalty functions. Adding penalty functions can often strengthen the robustness of the algorithm, encourages sparsity and incorporates prior knowledge into the problem [Hastie et al., 2001].

## 2.2 Coordinate Descent Optimization

Coordinate descent (CD)[1] and the more general block coordinate descent (BCD) algorithms solve a minimization problem by iteratively updating variables along coordinate directions or coordinate hyperplanes [Wright, 2015]. At each iteration of BCD, the objective function is (approximately) minimized w.r.t. a coordinate or a block of coordinates by fixing the remaining ones, and thus, an easier lower-dimensional subproblem needs to be solved. A number of comprehensive studies on BCD have already been carried out, such as Luo and Tseng [1992] and Nesterov [2012] for convex problems, and Tseng [2001], Xu and Yin [2013], and Razaviyayn et al. [2013] for nonconvex cases (also see Wright 2015 for a review paper). For stochastic problems with a block structure, Dang and Lan [2015] proposed stochastic block mirror descent (SBMD) by combining BCD with stochastic mirror descent [Beck and Teboulle, 2003; Nemirovski et al., 2009]. Another line of research on this topic is block stochastic gradient coordinate descent (BSG) [Xu and Yin, 2015]. The key difference between SBMD and BSG is that at each iteration, SBMD randomly picks one block of variables to update, while BSG cyclically updates all block variables.

In this paper, we develop mean-variance optimization algorithms based on both nonconvex stochastic BSG and SBMD. Since it has been shown that the BSG-based methods usually have better empirical performance than their SBMD counterparts, the main algorithm we report, analyze, and evaluate in the paper is BSG-based. We report our SBMD-based algorithm in Appendix C and use it as a baseline in the experiments of Section 5. The finite-sample analysis of our BSG-based algorithm reported in Section 4 is novel because although there exists such analysis for convex stochastic BSG methods [Xu and Yin, 2015], we are not aware of similar results for their nonconvex version to the best our knowledge.

## 3 Algorithm Design

In this section, we first discuss the difficulties of using the regular stochastic gradient ascent to maximize the mean-variance objective function. We then propose a new formulation of the mean-

variance objective function that is based on its Legendre-Fenchel dual and derive novel algorithms that are based on the recent results in stochastic nonconvex block coordinate descent. We conclude this section with an asymptotic analysis of a version of our proposed algorithm.

## 3.1 Problem Formulation

In this section, we describe why the vanilla stochastic gradient cannot be used to maximize $J_\lambda(\theta)$ defined in Eq. (2). Taking the gradient of $J_\lambda(\theta)$ w.r.t. $\theta$, we have

$$
\begin{aligned}
\nabla_\theta J_\lambda(\theta_t) =& \nabla_\theta J(\theta_t) - \lambda \nabla_\theta \mathrm{Var}(R) \\
=& \nabla_\theta J(\theta_t) - \lambda \big( \nabla_\theta M(\theta) - 2J(\theta)\nabla_\theta J(\theta) \big).
\end{aligned}
\tag{3}
$$

Computing $\nabla_\theta J_\lambda(\theta_t)$ in (3) involves computing three quantities: $\nabla_\theta J(\theta), \nabla_\theta M(\theta)$, and $J(\theta)\nabla_\theta J(\theta)$. We can obtain unbiased estimates of $\nabla_\theta J(\theta)$ and $\nabla_\theta M(\theta)$ from a single trajectory generated by the policy $\pi_\theta$ using the likelihood ratio method [Williams, 1992], as $\nabla_\theta J(\theta) = \mathbb{E}[R_t\omega_t(\theta)]$ and $\nabla_\theta M(\theta) = \mathbb{E}[R_t^2\omega_t(\theta)]$. Note that $R_t$ is the cumulative reward of the $t$-th episode, i.e., $R_t = \sum_{k=1}^{\tau_t} r_k$, which is possibly a *nonconvex* function, and $\omega_t(\theta) = \sum_{k=1}^{\tau_t} \nabla_\theta \ln \pi_\theta(a_k|s_k)$ is the likelihood ratio derivative. In the setting considered in the paper, an episode is the trajectory between two visits to the recurrent state $s^*$. For example, the $t$-th episode refers to the trajectory between the $(t\text{-}1)$-th and the $t$-th visits to $s^*$. We denote by $\tau_t$ the length of this episode.

However, it is not possible to compute an unbiased estimate of $J(\theta)\nabla_\theta J(\theta)$ without having access to a generative model of the environment that allows us to sample at least two next states $s'$ for each state-action pair $(s, a)$. As also noted by Tamar et al. [2012] and Prashanth and Ghavamzadeh [2013], computing an unbiased estimate of $J(\theta)\nabla_\theta J(\theta)$ requires double sampling (sampling from two different trajectories), and thus, cannot be done using a single trajectory. To circumvent the double-sampling problem, these papers proposed multi-time-scale stochastic approximation algorithms, the former a policy gradient algorithm and the latter an actor-critic algorithm that uses simultaneous perturbation methods [Bhatnagar et al., 2013]. However, as discussed in Section 2.1, multi-time-scale stochastic approximation approach suffers from several weaknesses such as no available finite-sample analysis and difficult-to-tune stepsize schedules. To overcome these weaknesses, we reformulate the mean-variance objective function and use it to present novel algorithms with in-depth analysis in the rest of the paper.

## 3.2 Block Coordinate Reformulation

In this section, we present a new formulation for $J_\lambda(\theta)$ that is later used to derive our algorithms and do not suffer from the double-sampling problem in estimating $J(\theta)\nabla_\theta J(\theta)$. We begin with the following lemma.

**Lemma 1.** *For the quadratic function $f(z) = z^2$, $z \in \mathbb{R}$, we define its Legendre-Fenchel dual as $f(z) = z^2 = \max_{y \in \mathbb{R}}(2zy - y^2)$.*

This is a special case of the Lengendre-Fenchel duality [Boyd and Vandenberghe, 2004] that has been used in several recent RL papers (e.g., Liu et al. 2015; Du et al. 2017; Liu et al. 2018). Let $F_\lambda(\theta) := \big(J(\theta) + \frac{1}{2\lambda}\big)^2 - M(\theta)$, which follows $F_\lambda(\theta) = \frac{J_\lambda(\theta)}{\lambda} + \frac{1}{4\lambda^2} - \zeta$. Since $\lambda > 0$ is a constant, maximizing $J_\lambda(\theta)$ is equivalent to maximizing $F_\lambda(\theta)$. Using Lemma 1 with $z = J(\theta) + \frac{1}{2\lambda}$, we may reformulate $F_\lambda(\theta)$ as

$$
F_\lambda(\theta) = \max_y \Big( 2y\big(J(\theta) + \frac{1}{2\lambda}\big) - y^2 \Big) - M(\theta).
\tag{4}
$$

Using (4), the maximization problem $\max_\theta F_\lambda(\theta)$ is equivalent to

$$
\begin{aligned}
&\max_{\theta, y} \quad \hat{f}_\lambda(\theta, y), \\
&\text{where} \quad \hat{f}_\lambda(\theta, y) := 2y\big(J(\theta) + \frac{1}{2\lambda}\big) - y^2 - M(\theta).
\end{aligned}
\tag{5}
$$

Our optimization problem is now formulated as the standard nonconvex coordinate ascent problem (5). We use three stochastic solvers to solve (5): SBMD method [Dang and Lan, 2015], BSG method [Xu and Yin, 2015], and the vanilla stochastic gradient ascent (SGA) method [Nemirovski et al., 2009]. We report our BSG-based algorithm in Section 3.3 and leave the details of the SBMD and SGA based algorithms to Appendix C. In the following sections, we denote by $\beta_t^\theta$ and $\beta_t^y$ the stepsizes of $\theta$ and $y$, respectively, and by the subscripts $t$ and $k$ the episode and time-step numbers.

### 3.3 Mean-Variance Policy Gradient

We now present our main algorithm that is based on a block coordinate update to maximize (5). Let $g_t^\theta$ and $g_t^y$ be block gradients and $\tilde{g}_t^\theta$ and $\tilde{g}_t^y$ be their sample-based estimations defined as

$$g_t^y = \mathbb{E}[\tilde{g}_t^y] = 2J(\theta_t) + \frac{1}{\lambda} - 2y_t \quad , \quad \tilde{g}_t^y = 2R_t + \frac{1}{\lambda} - 2y_t, \tag{6}$$

$$g_t^\theta = \mathbb{E}[\tilde{g}_t^\theta] = 2y_{t+1}\nabla_\theta J(\theta_t) - \nabla_\theta M(\theta_t) \quad , \quad \tilde{g}_t^\theta = \left(2y_{t+1}R_t - (R_t)^2\right)\omega_t(\theta_t). \tag{7}$$

The block coordinate updates are

$$y_{t+1} = y_t + \beta_t^y \tilde{g}_t^y,$$
$$\theta_{t+1} = \theta_t + \beta_t^\theta \tilde{g}_t^\theta.$$

To obtain unbiased estimates of $g_t^y$ and $g_t^\theta$, we shall update $y$ (to obtain $y_{t+1}$) prior to computing $g_t^\theta$ at each iteration. Now it is ready to introduce the Mean-Variance Policy Gradient (**MVP**) Algorithm 1. Before presenting our theoretical analysis, we first introduce the assumptions needed for these results.

---

**Algorithm 1** Mean-Variance Policy Gradient (**MVP**)

---

1: **Input:** Stepsizes $\{\beta_t^\theta\}$ and $\{\beta_t^y\}$, and number of iterations $N$
    **Option I:** $\{\beta_t^\theta\}$ and $\{\beta_t^y\}$ satisfy the Robbins-Monro condition
    **Option II:** $\beta_t^\theta$ and $\beta_t^y$ are set to be constants
2: **for** episode $t = 1, \ldots, N$ **do**
3:     Generate the initial state $s_1 \sim P_0$
4:     **while** $s_k \neq s^*$ **do**
5:         Take the action $a_k \sim \pi_{\theta_t}(a|s_k)$ and observe the reward $r_k$ and next state $s_{k+1}$
6:     **end while**
7:     Update the parameters

$$R_t = \sum_{k=1}^{\tau_t} r_k$$

$$\omega_t(\theta_t) = \sum_{k=1}^{\tau_t} \nabla_\theta \ln \pi_{\theta_t}(a_k|s_k)$$

$$y_{t+1} = y_t + \beta_t^y \left(2R_t + \frac{1}{\lambda} - 2y_t\right)$$

$$\theta_{t+1} = \theta_t + \beta_t^\theta \left(2y_{t+1}R_t - (R_t)^2\right)\omega_t(\theta_t)$$

8: **end for**
9: **Output** $\bar{x}_N$:
    **Option I:** Set $\bar{x}_N = x_N = [\theta_N, y_N]^\top$
    **Option II:** Set $\bar{x}_N = x_z = [\theta_z, y_z]^\top$, where $z$ is uniformly drawn from $\{1, 2, \ldots, N\}$

---

**Assumption 1** (**Bounded Gradient and Variance**). *There exist constants $G$ and $\sigma$ such that*

$$\|\nabla_y \hat{f}_\lambda(x)\|_2 \leq G, \ \|\nabla_\theta \hat{f}_\lambda(x)\|_2 \leq G,$$
$$\mathbb{E}[\|\Delta_t^y\|_2^2] \leq \sigma^2, \ \mathbb{E}[\|\Delta_t^\theta\|_2^2] \leq \sigma^2,$$

*for any $t$ and $x$, where $\|\cdot\|_2$ denotes the Euclidean norm, $\Delta_t^y := \tilde{g}_t^y - g_t^y$ and $\Delta_t^\theta := \tilde{g}_t^\theta - g_t^\theta$.*

Assumption 1 is standard in nonconvex coordinate descent algorithms [Xu and Yin, 2015; Dang and Lan, 2015]. We also need the following assumption that is standard in the policy gradient literature.

**Assumption 2** (**Ergodicity**). *The Markov chains induced by all the policies generated by the algorithm are ergodic, i.e., irreducible, aperiodic, and recurrent.*

In practice, we can choose either Option I with the result of the final iteration as output or Option II with the result of a randomly selected iteration as output. In what follows in this section, we report an

asymptotic convergence analysis of MVP with Option I, and in Section 4, we derive a finite-sample analysis of MVP with Option II.

**Theorem 1 (Asymptotic Convergence).** *Let $\{x_t = (\theta_t, y_t)\}$ be the sequence of the outputs generated by Algorithm 1 with Option I. If $\{\beta_t^\theta\}$ and $\{\beta_t^y\}$ are time-diminishing real positive sequences satisfying the Robbins-Monro condition, i.e., $\sum_{t=1}^\infty \beta_t^\theta = \infty$, $\sum_{t=1}^\infty (\beta_t^\theta)^2 < \infty$, $\sum_{t=1}^\infty \beta_t^y = \infty$, and $\sum_{t=1}^\infty (\beta_t^y)^2 < \infty$, then Algorithm 1 will converge such that $\lim_{t\to\infty} \mathbb{E}[\|\nabla \hat{f}_\lambda(x_t)\|_2] = 0$.*

The proof of Theorem 1 follows from the analysis in Xu and Yin [2013]. Due to space constraint, we report it in Appendix A.

Algorithm 1 is a special case of nonconvex block stochastic gradient (BSG) methods. To the best of our knowledge, no finite-sample analysis has been reported for this class of algorithms. Motivated by the recent papers by Nemirovski et al. [2009], Ghadimi and Lan [2013], Xu and Yin [2015], and Dang and Lan [2015], in Section 4, we provide a finite-sample analysis for general nonconvex block stochastic gradient methods and apply it to Algorithm 1 with Option II.

## 4 Finite-Sample Analysis

In this section, we first present a finite-sample analysis for the general class of nonconvex BSG algorithms [Xu and Yin, 2013], for which there are no established results, in Section 4.1. We then use these results and prove a finite-sample bound for our MVP algorithm with Option II, that belongs to this class, in Section 4.2. Due to space constraint, we report the detailed proofs in Appendix A.

### 4.1 Finite-Sample Analysis of Nonconvex BSG Algorithms

In this section, we provide a finite-sample analysis of the general nonconvex block stochastic gradient (BSG) method, where the problem formulation is given by

$$\min_{x\in\mathbb{R}^n} f(x) = \mathbb{E}_\xi[F(x,\xi)]. \tag{8}$$

$\xi$ is a random vector, and $F(\cdot,\xi): \mathbb{R}^n \to \mathbb{R}$ is continuously differentiable and possibly nonconvex for every $\xi$. The variable $x \in \mathbb{R}^n$ can be partitioned into $b$ disjoint blocks as $x = \{x^1, x^2, \ldots, x^b\}$, where $x^i \in \mathbb{R}^{n_i}$ denotes the $i$-th block of variables, and $\sum_{i=1}^b n_i = n$. For simplicity, we use $x^{<i}$ for $(x_i, \ldots, x_{i-1})$, and $x^{\leq i}, x^{>i}$, and $x^{\geq i}$ are defined correspondingly. We also use $\nabla_{x^i}$ to denote $\frac{\partial}{\partial x^i}$ for the partial gradient with respect to $x^i$. $\Xi_t$ is the sample set generated at $t$-th iteration, and $\mathbf{\Xi}_{[t]} = (\Xi_1, \ldots, \Xi_t)$ denotes the history of sample sets from the first through $t$-th iteration. $\{\beta_t^i : i = 1, \cdots, b\}_{t=1}^\infty$ are denoted as the stepsizes. Also, let $\beta_t^{\max} = \max_i \beta_t^i$, and $\beta_t^{\min} = \min_i \beta_t^i$. Similar to Algorithm 1, the BSG algorithm cyclically updates all blocks of variables in each iteration, and the detailed algorithm for BSG method is presented in Appendix B.

Without loss of generality, we assume a fixed update order in the BSG algorithm. Let $\Xi_t = \{\xi_{t,1}, \ldots, \xi_{t,m_t}\}$ be the samples in the $t$-th iteration with size $m_t \geq 1$. Therefore, the stochastic partial gradient is computed as $\tilde{g}_t^i = \frac{1}{m_t} \sum_{l=1}^{m_t} \nabla_{x^i} F(x_{t+1}^{<i}, x_t^{\geq i}; \xi_{t,l})$. Similar to Section 3, we define $g_t^i = \nabla_{x^i} f(x_{t+1}^{<i}, x_t^{\geq i})$, and the approximation error as $\Delta_t^i = \tilde{g}_t^i - g_t^i$. We assume that the objective function $f$ is bounded and Lipschitz smooth, i.e., there exists a positive Lipschitz constant $L > 0$ such that $\|\nabla_{x^i} f(x) - \nabla_{x^i} f(y)\|_2 \leq L\|x - y\|_2$, $\forall i \in \{1, \ldots, b\}$ and $\forall x, y \in \mathbb{R}^n$. Each block gradient of $f$ is also bounded, i.e., there exist a positive constant $G$ such that $\|\nabla_{x^i} f(x)\|_2 \leq G$, for any $i \in \{1, \ldots, b\}$ and any $x \in \mathbb{R}^n$. We also need Assumption 1 for all block variables, i.e., $\mathbb{E}[\|\Delta_t^i\|_2] \leq \sigma$, for any $i$ and $t$. Then we have the following lemma.

**Lemma 2.** *For any $i$ and $t$, there exist a positive constant $A$, such that*

$$\|\mathbb{E}[\Delta_t^i | \mathbf{\Xi}_{[t-1]}]\|_2 \leq A\beta_t^{\max}. \tag{9}$$

The proof of Lemma 2 is in Appendix B. It should be noted that in practice, it is natural to take the final iteration's result as the output as in Algorithm 1. However, a standard strategy for analyzing nonconvex optimization methods is to pick up one previous iteration's result randomly according to a discrete probability distribution over $\{1, 2, \ldots, N\}$ [Nemirovski et al., 2009; Ghadimi and Lan, 2013;

Dang and Lan, 2015]. Similarly, our finite-sample analysis is based on the strategy that randomly pick up $\bar{x}_N = x_z$ according to

$$\Pr(z = t) = \frac{\beta_t^{\min} - \frac{L}{2}(\beta_t^{\max})^2}{\sum_{t=1}^{N}(\beta_t^{\min} - \frac{L}{2}(\beta_t^{\max})^2)}, \quad t = 1, \ldots, N. \tag{10}$$

Now we provide the finite-sample analysis result for the general nonconvex BSG algorithm as in [Xu and Yin, 2015].

**Theorem 2.** *Let the output of the nonconvex BSG algorithm be $\bar{x}_N = x_z$ according to Eq. (10). If stepsizes satisfy $2\beta_t^{\min} > L(\beta_t^{\max})^2$ for $t = 1, \cdots, N$, then we have*

$$\mathbb{E}\left[\|\nabla f(\bar{x}_N)\|_2^2\right] \leq \frac{f(x_1) - f^* + \sum_{t=1}^{N}(\beta_t^{\max})^2 C_t}{\sum_{t=1}^{N}(\beta_t^{\min} - \frac{L}{2}(\beta_t^{\max})^2)}, \tag{11}$$

*where $f^* = \min_x f(x)$. $C_t = (1 - \frac{L}{2}\beta_t^{\max})\sum_{i=1}^{b} L\sqrt{\sum_{j<i}(G^2 + \sigma^2)} + b\left(AG + \frac{L}{2}\sigma^2\right)$, where $G$ is the gradient bound, $L$ is the Lipschitz constants, $\sigma$ is the variance bound, and $A$ is defined in Eq. (9).*

As a special case, we discuss the convergence rate with constant stepsizes $\mathcal{O}(1/\sqrt{N})$ in Corollary 1, which implies that the sample complexity $N = \mathcal{O}(1/\varepsilon^2)$ in order to find $\varepsilon$-stationary solution of problem (8).

**Corollary 1.** *If we take constant stepsize such that $\beta_t^i = \beta^i = \mathcal{O}(1/\sqrt{N})$ for any $t$, and let $\beta^{\max} := \max_i \beta^i$, $\beta^{\min} := \min_i \beta^i$, then we have $\mathbb{E}\left[\|\nabla f(\bar{x}_N)\|_2^2\right] \leq \mathcal{O}\left(\sqrt{\frac{f(x_1) - f^* + C}{N}}\right)$, where $C_t$ in Eq. (11) reduces to a constant $C$ defined as $C = (1 - \frac{L}{2}\beta^{\max})\sum_{i=1}^{b} L\sqrt{\sum_{j<i}(G^2 + \sigma^2)} + b\left(AG + \frac{L}{2}\sigma^2\right)$.*

## 4.2 Finite-Sample Analysis of Algorithm 1

We present the major theoretical results of this paper, i.e., the finite-sample analysis of Algorithm 1 with Option II. The proof of Theorem 3 is in Appendix A.

**Theorem 3.** *Let the output of the Algorithm 1 be $\bar{x}_N$ as in Theorem 2. If $\{\beta_t^\theta\}$, $\{\beta_t^y\}$ are constants as in Option II in Algorithm 1, and also satisfies $2\beta_t^{\min} > L(\beta_t^{\max})^2$ for $t = 1, \cdots, N$, we have*

$$\mathbb{E}\left[\|\nabla \hat{f}_\lambda(\bar{x}_N)\|_2^2\right] \leq \frac{\hat{f}_\lambda^* - \hat{f}_\lambda(x_1) + N(\beta_t^{\max})^2 C}{N(\beta_t^{\min} - \frac{L}{2}(\beta_t^{\max})^2)} \tag{12}$$

*where $\hat{f}_\lambda^* = \max_x \hat{f}_\lambda(x)$, and*

$$C = (1 - \frac{L}{2}\beta_t^{\max})(L^2\beta_t^{\max}(G^2 + \sigma^2) + L(2G^2 + \sigma^2)) + AG + L\sigma^2 + 2L(1 + L\beta_t^{\max})(3\sigma^2 + 2G^2).$$

*Proof Sketch.* The proof follows the following major steps.

**(I).** First, we need to prove the bound of each block coordinate gradient, i.e., $\mathbb{E}[\|g_t^\theta\|_2^2]$ and $\mathbb{E}[\|g_t^y\|_2^2]$, which is bounded as

$$(\beta_t^{\min} - \frac{L}{2}(\beta_t^{\max})^2)\mathbb{E}[\|g_t^\theta\|_2^2 + \|g_t^y\|_2^2]$$

$$\leq \mathbb{E}[\hat{f}_\lambda(x_{t+1})] - \mathbb{E}[\hat{f}_\lambda(x_t)] + (\beta_t^{\max})^2 AM_\rho + L(\beta_t^{\max})^2\sigma^2 + 2L\beta_t^{\max}(\beta_t^{\max} + L(\beta_t^{\max})^2)(3\sigma^2 + 2G^2).$$

Summing up over $t$, we have

$$\sum_{t=1}^{N}(\beta_t^{\min} - \frac{L}{2}(\beta_t^{\max})^2)\mathbb{E}[\|g_t^\theta\|_2^2 + \|g_t^y\|_2^2]$$

$$\leq \hat{f}_\lambda^* - \hat{f}_\lambda(x_1) + \sum_{t=1}^{N}[(\beta_t^{\max})^2 AG + L(\beta_t^{\max})^2\sigma^2 + 2L\beta_t^{\max}(\beta_t^{\max} + L(\beta_t^{\max})^2)(3\sigma^2 + 2G^2)].$$

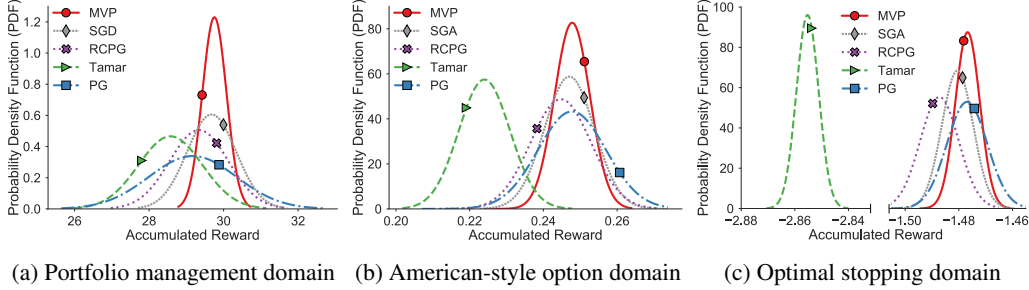

(a) Portfolio management domain    (b) American-style option domain    (c) Optimal stopping domain

Figure 1: Empirical results of the distributions of the return (cumulative rewards) random variable. Note that markers only indicate different methods.

**(II)**. Next, we need to bound $\mathbb{E}[\|\nabla \hat{f}_\lambda(x_t)\|_2^2]$ using $\mathbb{E}[\|g_t^\theta\|_2^2 + \|g_t^y\|_2^2]$, which is proven to be

$$\mathbb{E}[\|\nabla \hat{f}_\lambda(x_t)\|_2^2] \leq L^2(\beta_t^{\max})^2(G^2 + \sigma^2) + L\beta_t^{\max}(2G^2 + \sigma^2) + \mathbb{E}[\|g_t^\theta\|_2^2 + \|g_t^y\|_2^2].$$

**(III)**. Finally, combining (I) and (II), and rearranging the terms, Eq. (12) can be obtained as a special case of Theorem 2, which completes the proof. □

## 5  Experimental Study

In this section, we evaluate our MVP algorithm with Option I in three risk-sensitive domains: the portfolio management [Tamar et al., 2012], the American-style option [Tamar et al., 2014], and the optimal stopping [Chow and Ghavamzadeh, 2014; Chow et al., 2018]. The baseline algorithms are the vanilla policy gradient (PG), the mean-variance policy gradient in Tamar et al. [2012], the stochastic gradient ascent (SGA) applied to our optimization problem (5), and the randomized coordinate ascent policy gradient (RCPG), i.e., the SBMD-based version of our algorithm. Details of SGA and RCPG can be found in Appendix C. For each algorithm, we optimize its Lagrangian parameter $\lambda$ by grid search and report the mean and variance of its return random variable as a Gaussian.[2] Since the algorithms presented in the paper (MVP and RCPG) are policy gradient, we only compare them with Monte-Carlo based policy gradient algorithms and do not use any actor-critic algorithms, such as those in Prashanth and Ghavamzadeh [2013] and TRPO [Schulman et al., 2015], in the experiments.

### 5.1  Portfolio Management

The portfolio domain Tamar et al. [2012] is composed of the liquid and non-liquid assets. A liquid asset has a fixed interest rate $r_l$ and can be sold at any time-step $k \leq \tau$. A non-liquid asset can be sold only after a fixed period of $W$ time-steps with a time-dependent interest rate $r_{\text{nl}}(k)$, which can take either $r_{\text{nl}}^{\text{low}}$ or $r_{\text{nl}}^{\text{high}}$, and the transition follows a switching probability $p_{\text{switch}}$. The non-liquid asset also suffers a default risk (i.e., not being paid) with a probability $p_{\text{risk}}$. All investments are in liquid assets at the initial time-step $k = 0$. At the $k$-th step, the state is denoted by $x(k) \in \mathbb{R}^{W+2}$, where $x_1 \in [0, 1]$ is the portion of the investment in liquid assets, $x_2, \cdots, x_{W+1} \in [0, 1]$ is the portion in non-liquid assets with time to maturity of $1, \cdots, W$ time-steps, respectively, and $x_{W+2}(k) = r_{\text{nl}}(k) - \mathbb{E}[r_{\text{nl}}(k)]$. The investor can choose to invest a fixed portion $\eta$ $(0 < \eta < 1)$ of his total available cash in the non-liquid asset or do nothing. More details about this domain can be found in Tamar et al. [2012]. Figure 1(a) shows the results of the algorithms. PG has a large variance and the Tamar's method has the lowest mean return. The results indicate that MVP yields a higher mean return with less variance compared to the competing algorithms.

### 5.2  American-style Option

An American-style option Tamar et al. [2014] is a contract that gives the buyer the right to buy or sell the asset at a strike price $W$ at or before the maturity time $\tau$. The initial price of the option is $x_0$, and the buyer has bought a put option with the strike price $W_{\text{put}} < x_0$ and a call option with the

strike price $W_{\text{call}} > x_0$. At the $k$-th step ($k \leq \tau$), the state is $\{x_k, k\}$, where $x_k$ is the current price of the option. The action $a_k$ is either executing the option or holding it. $x_{k+1}$ is $f_u x_k$ w.p. $p$ and $f_d x_k$ w.p. $1 - p$, where $f_u$ and $f_d$ are constants. The reward is $0$ unless an option is executed and the reward for executing an option is $r_k = \max(0, W_{\text{put}} - x_k) + \max(0, x_k - W_{\text{call}})$. More details about this domain can be found in Tamar et al. [2014]. Figure 1(b) shows the performance of the algorithms. The results suggest that MVP can yield a higher mean return with less variance compared to the other algorithms.

### 5.3 Optimal Stopping

The optimal stopping problem [Chow and Ghavamzadeh, 2014; Chow et al., 2018] is a continuous state domain. At the $k$-th time-step ($k \leq \tau$, $\tau$ is the stopping time), the state is $\{x_k, k\}$, where $x_k$ is the cost. The buyer decide either to accept the present cost or wait. If the buyer accepts or when $k = T$, the system reaches a terminal state and the cost $x_k$ is received, otherwise, the buyer receives the cost $p_h$ and the new state is $\{x_{k+1}, k + 1\}$, where $x_{k+1}$ is $f_u x_k$ w.p. $p$ and $f_d x_k$ w.p. $1 - p$ ($f_u > 1$ and $f_d < 1$ are constants). More details about this domain can be found in Chow and Ghavamzadeh [2014]. Figure 1(c) shows the performance of the algorithms. The results indicate that MVP is able to yield much less variance without affecting its mean return. We also summarize the performance of these algorithms in all three risk-sensitive domains as Table 1, where Std is short for Standard Deviation.

| | Portfolio Management | | American-style Option | | Optimal Stopping | |
|---|---|---|---|---|---|---|
| | Mean | Std | Mean | Std | Mean | Std |
| MVP | **29.754** | **0.325** | **0.2478** | **0.00482** | **-1.4767** | 0.00456 |
| PG | 29.170 | 1.177 | 0.2477 | 0.00922 | -1.4769 | 0.00754 |
| Tamar | 28.575 | 0.857 | 0.2240 | 0.00694 | -2.8553 | **0.00415** |
| SGA | 29.679 | 0.658 | 0.2470 | 0.00679 | -1.4805 | 0.00583 |
| RCPG | 29.340 | 0.789 | 0.2447 | 0.00819 | -1.4872 | 0.00721 |

Table 1: Performance Comparison among Algorithms

## 6 Conclusion

This paper is motivated to provide a risk-sensitive policy search algorithm with provable sample complexity analysis to maximize the mean-variance objective function. To this end, the objective function is reformulated based on the Legendre-Fenchel duality, and a novel stochastic block coordinate ascent algorithm is proposed with in-depth analysis. There are many interesting future directions on this research topic. Besides stochastic policy gradient, deterministic policy gradient [Silver et al., 2014] has shown great potential in large discrete action space. It is interesting to design a risk-sensitive deterministic policy gradient method. Secondly, other reformulations of the mean-variance objective function are also worth exploring, which will lead to new families of algorithms. Thirdly, distributional RL [Bellemare et al., 2016] is strongly related to risk-sensitive policy search, and it is interesting to investigate the connections between risk-sensitive policy gradient methods and distributional RL. Last but not least, it is interesting to test the performance of the proposed algorithms together with other risk-sensitive RL algorithms on highly-complex risk-sensitive tasks, such as autonomous driving problems and other challenging tasks.

## Acknowledgments

Bo Liu, Daoming Lyu, and Daesub Yoon were partially supported by a grant (18TLRP-B131486-02) from Transportation and Logistics R&D Program funded by Ministry of Land, Infrastructure and Transport of Korean government. Yangyang Xu was partially supported by the NSF grant DMS-1719549.

## Footnotes

[1] Note that since our problem is maximization, our proposed algorithms are block coordinate *ascent*.

[2]Note that the return random variables are not necessarily Gaussian, we only use Gaussian for presentation purposes.

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
