[Supplementary Material]

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

# Appendix

## A   Theoretical Analysis of Algorithm 1

Now we present the theoretical analysis of Algorithm 1, with both asymptotic convergence and finite-sample error bound analysis. For the purpose of clarity, in the following analysis, $x$ is defined as $x := [y, \theta]^\top$, where $x^1 := y$, $x^2 := \theta$. Similarly, $g_t^i$ (resp. $\beta_t^i$) is used to denote $g_t^y$(resp. $\beta_t^y$) and $g_t^\theta$(resp. $\beta_t^\theta$), where $g_t^1 := g_t^y$, $g_t^2 := g_t^\theta$(resp. $\beta_t^1 := \beta_t^y$, $\beta_t^2 := \beta_t^\theta$). Let $\beta_t^{\max} = \max\{\beta_t^1, \beta_t^2\}$, $\beta_t^{\min} = \min\{\beta_t^1, \beta_t^2\}$, and $\|\cdot\|_2$ denotes the Euclidean norm.

### A.1   Asymptotic Convergence Proof Algorithm 1

In this section, we provide the asymptotic convergence analysis for Algorithm 1 with Option I, i.e., the stepsizes are chosen to satisfy the Robbins-Monro condition and the output is the last iteration's result. We first introduce a useful lemma, which follows from Lemma 2.

**Lemma 3.** *Under Assumption 1, we have*

$$\mathbb{E}[\langle \nabla_y \hat{f}_\lambda(x_t), \Delta_t^y \rangle] = 0$$
$$\mathbb{E}[\langle \nabla_\theta \hat{f}_\lambda(x_t), \Delta_t^\theta \rangle] \leq \beta_t^{\max} AG,$$

*where A is defined in Eq (9).*

*Proof.* Let $\Xi_t$ denote the history of random samples from the first to the $t$–th episode, then $x_t$ is independent of $\Delta_t^i$ conditioned on $\Xi_{t-1}$ (since $x_t$ is deterministic conditioned on $\Xi_{t-1}$). Then we have

$$\mathbb{E}[\langle \nabla_y \hat{f}_\lambda(x_t), \Delta_t^y \rangle] = \mathbb{E}_{\Xi_{t-1}}[\mathbb{E}[\langle \nabla_y \hat{f}_\lambda(x_t), \Delta_t^y \rangle | \Xi_{t-1}]]$$
$$= \mathbb{E}_{\Xi_{t-1}}[\langle \mathbb{E}[\nabla_y \hat{f}_\lambda(x_t) | \Xi_{t-1}], \mathbb{E}[\Delta_t^y | \Xi_{t-1}] \rangle] = 0, \tag{13}$$

$$\mathbb{E}[\langle \nabla_\theta \hat{f}_\lambda(x_t), \Delta_t^\theta \rangle] = \mathbb{E}_{\Xi_{t-1}}[\mathbb{E}[\langle \nabla_\theta \hat{f}_\lambda(x_t), \Delta_t^\theta \rangle | \Xi_{t-1}]]$$
$$= \mathbb{E}_{\Xi_{t-1}}[\langle \mathbb{E}[\nabla_\theta \hat{f}_\lambda(x_t) | \Xi_{t-1}], \mathbb{E}[\Delta_t^\theta | \Xi_{t-1}] \rangle]$$
$$\leq \mathbb{E}_{\Xi_{t-1}}[\|\mathbb{E}[\nabla_\theta \hat{f}_\lambda(x_t) | \Xi_{t-1}]\|_2 \cdot \|\mathbb{E}[\Delta_t^\theta | \Xi_{t-1}]\|_2]$$
$$\leq \beta_t^{\max} A \mathbb{E}_{\Xi_{t-1}}[\|\mathbb{E}[\nabla_\theta \hat{f}_\lambda(x_t) | \Xi_{t-1}]\|_2]$$
$$\leq \beta_t^{\max} A \mathbb{E}[\|\nabla_\theta \hat{f}_\lambda(x_t)\|_2] \leq \beta_t^{\max} AG, \tag{14}$$

where the second equality in both Eq.(13) and Eq.(14) follows from the conditional independence between $x_t$ and $\Delta_t^i$, the second inequality in Eq.(14) follows from Assumption 1, and the last inequality in Eq.(14) follows from the Jensen's inequality and $G$ is the gradient bound.    □

Then, we introduce Lemma 4, which is essential for proving Theorem 1.

**Lemma 4.** *For two non-negative scalar sequences $\{a_t\}$ and $\{b_t\}$, if $\sum_{t=1}^\infty a_t = +\infty$ and $\sum_{t=1}^\infty a_t b_t < +\infty$, we then have*

$$\lim_{t\to\infty} \inf b_t = 0.$$

*Further, if there exists a constant $K > 0$ such that $|b_{t+1} - b_t| \leq a_t K$, then*

$$\lim_{t\to\infty} b_t = 0.$$

The detailed proof can be found in Lemma A.5 of [Mairal, 2013] and Proposition 1.2.4 of [Bertsekas, 1999]. Now it is ready to prove Theorem 1.

*Proof of Theorem 1.* We define $\Gamma_t^1 := \hat{f}(x_t^1, x_t^2) - \hat{f}(x_{t+1}^1, x_t^2), \Gamma_t^2 := \hat{f}(x_{t+1}^1, x_t^2) - \hat{f}(x_{t+1}^1, x_{t+1}^2)$ to denote the block update. It turns out that for $i = 1, 2$, and $\Gamma_t^i$ can be bounded following the

Lipschitz smoothness as

$$
\begin{aligned}
\Gamma_t^i \leq & \langle g_t^i, x_t^i - x_{t+1}^i \rangle + \frac{L}{2}\|x_t^i - x_{t+1}^i\|_2^2 \\
= & -\beta_t^i \langle g_t^i, \tilde{g}_t^i \rangle + \frac{L}{2}(\beta_t^i)^2 \|\tilde{g}_t^i\|_2^2 \\
= & -(\beta_t^i - \frac{L}{2}(\beta_t^i)^2)\|g_t^i\|_2^2 + \frac{L}{2}(\beta_t^i)^2)\|\Delta_t^i\|_2^2 - (\beta_t^i - L(\beta_t^i)^2)\langle g_t^i, \Delta_t^i \rangle \\
= & -(\beta_t^i - \frac{L}{2}(\beta_t^i)^2)\|g_t^i\|_2^2 + \frac{L}{2}(\beta_t^i)^2\|\Delta_t^i\|_2^2 \\
& -(\beta_t^i - L(\beta_t^i)^2)(\langle g_t^i - \nabla_{x^i}\hat{f}_\lambda(x_t), \Delta_t^i \rangle + \langle \nabla_{x^i}\hat{f}_\lambda(x_t), \Delta_t^i \rangle)
\end{aligned}
\tag{15}
$$

where the equalities follow the definition of $\Delta_t^i$ and the update law of Algorithm 1. We also have the following argument

$$
\begin{aligned}
& -(\beta_t^i - L(\beta_t^i)^2)\langle g_t^i - \nabla_{x^i}\hat{f}_\lambda(x_t), \Delta_t^i \rangle \\
\leq & |\beta_t^i - L(\beta_t^i)^2|\|\Delta_t^i\|_2\|g_t^i - \nabla_{x^i}\hat{f}_\lambda(x_t)\|_2 \\
\leq & L|\beta_t^i - L(\beta_t^i)^2|\|\Delta_t^i\|_2\|x_{t+1} - x_t\|_2 \\
\leq & L|\beta_t^i - L(\beta_t^i)^2|\|\Delta_t^i\|_2 \sqrt{\sum_{j=1}^{2}\|\beta_t^j \tilde{g}_t^j\|_2^2} \\
\leq & L(\beta_t^i + L(\beta_t^i)^2)\beta_t^{\max}\left(\|\Delta_t^i\|_2 + \sum_{j=1}^{2}(\|g_t^j\|_2^2 + \|\Delta_t^j\|_2^2)\right),
\end{aligned}
\tag{16}
$$

where the first inequality follows from Cauchy-Schwarz inequality, the second inequality follows from the Lipschitz smoothness of objective function $\hat{f}_\lambda$, the third inequality follows from the update law of Algorithm 1, and the last inequality follows from the triangle inequality. Combining Eq. (15) and Eq. (16), we obtain

$$
\begin{aligned}
\Gamma_t^i \leq & -(\beta_t^i - \frac{L}{2}(\beta_t^i)^2)\|g_t^i\|_2^2 + \frac{L}{2}(\beta_t^i)^2\|\Delta_t^i\|_2^2 \\
& -(\beta_t^i - L(\beta_t^i)^2)\langle \nabla_{x^i}\hat{f}_\lambda(x_t), \Delta_t^i \rangle \\
& + L(\beta_t^i + L(\beta_t^i)^2) \cdot \beta_t^{\max}\left(\|\Delta_t^i\|_2^2 + \sum_{j=1}^{2}(\|g_t^j\|_2^2 + \|\Delta_t^j\|_2^2)\right).
\end{aligned}
\tag{17}
$$

Summing Eq. (17) over $i$, then we obtain

$$
\hat{f}_\lambda(x_t) - \hat{f}_\lambda(x_{t+1})
\tag{18}
$$
$$
\begin{aligned}
\leq & -\sum_{i=1}^{2}(\beta_t^i - \frac{L}{2}(\beta_t^i)^2)\|g_t^i\|_2^2 - \sum_{i=1}^{2}(\beta_t^i - L(\beta_t^i)^2)\langle \nabla_{x^i}\hat{f}_\lambda(x_t), \Delta_t^i \rangle \\
& + \sum_{i=1}^{2}\left(\frac{L}{2}(\beta_t^i)^2\|\Delta_t^i\|_2^2 + L(\beta_t^i + L(\beta_t^i)^2)\beta_t^{\max}(\|\Delta_t^i\|_2^2 + \sum_{j=1}^{2}(\|g_t^j\|_2^2 + \|\Delta_t^j\|_2^2))\right).
\end{aligned}
$$

We also have the following fact,

$$
\begin{aligned}
\mathbb{E}[\langle \nabla_y \hat{f}_\lambda(x_t), \Delta_t^y \rangle] = & 0 \\
\mathbb{E}[\langle \nabla_\theta \hat{f}_\lambda(x_t), \Delta_t^\theta \rangle] \leq & \beta_t^{\max}AG.
\end{aligned}
$$

We prove this fact in Lemma 3 as a special case of Lemma 2, and the general analysis can be found in Lemma 1 in [Xu and Yin, 2015]. Taking expectation w.r.t. $t$ on both sides of the inequality Eq. (18),

we have

$$
\mathbb{E}[\hat{f}_\lambda(x_t)] - \mathbb{E}[\hat{f}_\lambda(x_{t+1})]
$$

$$
\leq -\sum_{i=1}^{2}(\beta_t^i - \frac{L}{2}(\beta_t^i)^2)\mathbb{E}[\|g_t^i\|_2^2] + (\beta_t^\theta - L(\beta_t^\theta)^2)\beta_t^{\max}AG
$$

$$
+ \sum_{i=1}^{2}\left(\frac{L}{2}(\beta_t^i)^2\mathbb{E}[\|\Delta_t^i\|_2^2] + L(\beta_t^i + L(\beta_t^i)^2)\beta_t^{\max}(\mathbb{E}[\|\Delta_t^i\|_2^2] + \sum_{j=1}^{2}(\mathbb{E}[\|g_t^j\|_2^2] + \mathbb{E}[\|\Delta_t^j\|_2^2]))\right)
$$

$$
\leq -\sum_{i=1}^{2}(\beta_t^{\min} - \frac{L}{2}(\beta_t^{\max})^2)\mathbb{E}[\|g_t^i\|_2^2] + (\beta_t^{\max})^2AG
$$

$$
+ (L(\beta_t^{\max})^2\sigma^2 + 2L\beta_t^{\max}(\beta_t^{\max} + L(\beta_t^{\max})^2)(3\sigma^2 + 2G^2)), \tag{19}
$$

where the first inequality follows from Eq. (13) and Eq. (14), and the second inequality follows from the boundedness of $\mathbb{E}[\|\Delta_t^i\|_2]$ and $\mathbb{E}[\|g_t^i\|_2]$.

Rearranging Eq. (19), we obtain

$$
\sum_{i=1}^{2}(\beta_t^{\min} - \frac{L}{2}(\beta_t^{\max})^2)\mathbb{E}[\|g_t^i\|_2^2]
$$

$$
\leq \mathbb{E}[\hat{f}_\lambda(x_{t+1})] - \mathbb{E}[\hat{f}_\lambda(x_t)] + (\beta_t^{\max})^2AM_\rho + L(\beta_t^{\max})^2\sigma^2
$$

$$
+ 2L\beta_t^{\max}(\beta_t^{\max} + L(\beta_t^{\max})^2)(3\sigma^2 + 2G^2). \tag{20}
$$

By further assuming $0 < \inf_t \frac{\{\beta_t^\theta\}}{\{\beta_t^y\}} \leq \sup_t \frac{\{\beta_t^\theta\}}{\{\beta_t^y\}} < \infty$, it can be verified that $\{\beta_t^{\max}\}$ and $\{\beta_t^{\min}\}$ also satisfy Robbins-Monro condition. Note that $\hat{f}_\lambda$ is upper bounded, summing Eq. (20) over $t$ and using the Robbins-Monro condition of $\{\beta_t^\theta\}, \{\beta_t^y\}, \{\beta_t^{\max}\}, \{\beta_t^{\min}\}$, we have

$$
\sum_{t=1}^{\infty}\beta_t^{\min}\mathbb{E}[\|g_t^i\|_2^2] < \infty, \ \forall i. \tag{21}
$$

Furthermore, let $\xi_t^1 = (x_t^1, x_t^2)$ and $\xi_t^2 = (x_{t+1}^1, x_t^2)$, then

$$
|\mathbb{E}[\|g_{t+1}^i\|_2^2] - \mathbb{E}[\|g_t^i\|_2^2]| \leq \mathbb{E}[\|g_{t+1}^i + g_t^i\|_2 \cdot \|g_{t+1}^i - g_t^i\|_2]
$$

$$
\leq 2LM_\rho\mathbb{E}[\|\xi_{t+1}^i - \xi_t^i\|_2]
$$

$$
= 2LM_\rho\mathbb{E}\left[\sqrt{\sum_{j<i}\|\beta_{t+1}^j\tilde{g}_{t+1}^j\|_2^2 + \sum_{j\geq i}\|\beta_t^j\tilde{g}_t^j\|_2^2}\right]
$$

$$
\leq 2LM_\rho\beta_t^{\max}\mathbb{E}\left[\sqrt{\sum_{j<i}\|\tilde{g}_{t+1}^j\|_2^2 + \sum_{j\geq i}\|\tilde{g}_t^j\|_2^2}\right]
$$

$$
\leq 2LM_\rho\beta_t^{\max}\sqrt{\mathbb{E}[\sum_{j<i}\|\tilde{g}_{t+1}^j\|_2^2 + \sum_{j\geq i}\|\tilde{g}_t^j\|_2^2]}
$$

$$
\leq 2LM_\rho\beta_t^{\max}\sqrt{2(G^2 + \sigma^2)}, \tag{22}
$$

where the first inequality follows from Jensen's inequality, the second inequality follows from the definition of gradient bound $G$ and the gradient Lipschitz continuity of $\hat{f}_\lambda$, the third inequality follows from the Robbins-Monro condition of $\{\beta_t^y\}$ and $\{\beta_t^\theta\}$, and the last two inequalities follow Jensen's inequality in probability theory.

Combining Eq. (21) and Eq. (22) and according to Lemma 4, we have $\lim_{t\to\infty}\mathbb{E}[\|g_t^i\|_2]=0$ for $i=1,2$ by Jensen's inequality. Hence,

$$\mathbb{E}[\|\nabla_y\hat{f}_\lambda(x_t)\|_2] = \mathbb{E}[\|g_t^y\|_2]$$

$$\mathbb{E}[\|\nabla_\theta\hat{f}_\lambda(x_t)\|_2] \leq \mathbb{E}[\|\nabla_\theta\hat{f}_\lambda(x_t)-g_t^\theta\|_2] + \mathbb{E}[\|g_t^\theta\|_2]$$

$$\leq L\cdot\mathbb{E}[\|y_{t+1}-y_t\|_2] + \mathbb{E}[\|g_t^\theta\|_2]$$

$$\leq L\beta_t^{\max}\mathbb{E}[\|\tilde{g}_t^y\|_2] + \mathbb{E}[\|g_t^\theta\|_2]$$

$$\leq L\beta_t^{\max}\sqrt{G^2+\sigma^2} + \mathbb{E}[\|g_t^\theta\|_2]$$

where the first inequality follows from the triangle inequality, the second inequality follows from the Lipschitz continuity of $\hat{f}_\lambda$, and the last inequality follows from the same argument for Eq. (22). Also, note that $\lim_{t\to\infty}\beta_t^{\max}=0$, $\lim_{t\to\infty}L\sqrt{G^2+\sigma^2}<\infty$, and $\lim_{t\to\infty}\mathbb{E}[\|g_t^i\|_2]=0$, so that when $t\to\infty$, $L\beta_t^{\max}\sqrt{G^2+\sigma^2}+\mathbb{E}[\|g_t^y\|_2]+\mathbb{E}[\|g_t^\theta\|_2]\to 0$. This completes the proof. □

**Remark 1.** *Different from the stringent two-time-scale setting where one stepsize needs to be "quasi-stationary" compared to the other [Tamar et al., 2012], the stepsizes in Algorithm 1 does not have such requirements, which makes it easy to tune in practice.*

## A.2 Finite-Sample Analysis of Algorithm 1

The above analysis provides asymptotic convergence guarantee of Algorithm 1, however, it is desirable to know the sample complexity of the algorithm in real applications. Motivated by offering RL practitioners confidence in applying the algorithm, we then present the sample complexity analysis with Option II described in Algorithm 1, i.e., the stepsizes are set to be a constant, and the output is randomly selected from $\{x_1,\cdots,x_N\}$ with a discrete uniform distribution. This is a standard strategy for nonconvex stochastic optimization approaches [Dang and Lan, 2015]. With these algorithmic refinements, we are ready to present the finite-sample analysis as follows. It should be noted that this proof is a special case of the general stochastic nonconvex BSG algorithm analysis provided in Appendix B.2.

*Proof of Theorem 3.* The proof of finite-sample analysis starts from the similar idea with asymptotic convergence. The following analysis follows from Eq. (20) in the proof of Theorem 1, but we are using stepsizes $\{\beta_t^\theta\}$, $\{\beta_t^y\}$ are constants which satisfy $2\beta_t^{\min}>L(\beta_t^{\max})^2$ for $t=1,\cdots,N$ in this proof.

Summing Eq. (20) over $t$, we have

$$\sum_{t=1}^{N}\sum_{i=1}^{2}(\beta_t^{\min}-\frac{L}{2}(\beta_t^{\max})^2)\mathbb{E}[\|g_t^i\|_2^2] \tag{23}$$

$$\leq \hat{f}_\lambda^* - \hat{f}_\lambda(x_1) + \sum_{t=1}^{N}[(\beta_t^{\max})^2 AG + L(\beta_t^{\max})^2\sigma^2 + 2L\beta_t^{\max}(\beta_t^{\max}+L(\beta_t^{\max})^2)(3\sigma^2+2G^2)].$$

Next, we bound $\mathbb{E}[\|\nabla\hat{f}_\lambda(x_t)\|_2^2]$ using $\mathbb{E}[\|g_t^\theta\|_2^2+\|g_t^y\|_2^2]$

$$\mathbb{E}[\|\nabla\hat{f}_\lambda(x_t)\|_2^2] = \mathbb{E}[\|\nabla_\theta\hat{f}_\lambda(x_t)\|_2^2+\|\nabla_y\hat{f}_\lambda(x_t)\|_2^2]$$

$$\leq \mathbb{E}[\|\nabla_\theta\hat{f}_\lambda(\theta_t,y_t)-\nabla_\theta\hat{f}_\lambda(\theta_t,y_{t+1})+\nabla_\theta\hat{f}_\lambda(\theta_t,y_{t+1})\|_2^2+\|g_t^y\|_2^2]$$

$$\leq \mathbb{E}[\|\nabla_\theta\hat{f}_\lambda(\theta_t,y_t)-g_t^\theta\|_2^2+\|g_t^\theta\|_2^2+2\langle\nabla_\theta\hat{f}_\lambda(\theta_t,y_t)-g_t^\theta,g_t^\theta\rangle+\|g_t^y\|_2^2]$$

$$\leq \mathbb{E}[\|\nabla_\theta\hat{f}_\lambda(\theta_t,y_t)-g_t^\theta\|_2^2+2\langle\nabla_\theta\hat{f}_\lambda(\theta_t,y_t)-g_t^\theta,g_t^\theta\rangle]+\mathbb{E}[\|g_t^\theta\|_2^2+\|g_t^y\|_2^2]$$

$$\leq \mathbb{E}[L^2\|y_{t+1}-y_t\|_2^2+2L\|y_{t+1}-y_t\|_2\cdot\|g_t^\theta\|_2]+\mathbb{E}[\|g_t^\theta\|_2^2+\|g_t^y\|_2^2]$$

$$\leq \mathbb{E}[L^2(\beta_t^y)^2\|\tilde{g}_t^y\|_2^2+2L\beta_t^y\|\tilde{g}_t^y\|_2\cdot\|g_t^\theta\|_2]+\mathbb{E}[\|g_t^\theta\|_2^2+\|g_t^y\|_2^2]$$

$$\leq L^2(\beta_t^y)^2(G^2+\sigma^2)+2L\beta_t^y\mathbb{E}[\|\tilde{g}_t^y\|_2\cdot\|g_t^\theta\|_2]+\mathbb{E}[\|g_t^\theta\|_2^2+\|g_t^y\|_2^2]$$

$$\leq L^2(\beta_t^y)^2(G^2+\sigma^2)+L\beta_t^y\mathbb{E}[\|\tilde{g}_t^y\|_2^2+\|g_t^\theta\|_2^2]+\mathbb{E}[\|g_t^\theta\|_2^2+\|g_t^y\|_2^2]$$

$$\leq L^2(\beta_t^{\max})^2(G^2+\sigma^2)+L\beta_t^{\max}(2G^2+\sigma^2)+\mathbb{E}[\|g_t^\theta\|_2^2+\|g_t^y\|_2^2]. \tag{24}$$

Then, combine Eq. (24) with Eq. (23)

$$\sum_{t=1}^{N}(\beta_t^{\min} - \frac{L}{2}(\beta_t^{\max})^2)\mathbb{E}[\|\nabla \hat{f}_\lambda(x_t)\|_2^2]$$

$$\leq \sum_{t=1}^{N}(\beta_t^{\min} - \frac{L}{2}(\beta_t^{\max})^2)\mathbb{E}[L^2(\beta_t^{\max})^2(G^2+\sigma^2) + L\beta_t^{\max}(2G^2+\sigma^2) + \|g_t^\theta\|_2^2 + \|g_t^y\|_2^2]$$

$$\leq \hat{f}_\lambda^* - \hat{f}_\lambda(x_1) + \sum_{t=1}^{N}[(\beta_t^{\min} - \frac{L}{2}(\beta_t^{\max})^2)(L^2(\beta_t^{\max})^2(G^2+\sigma^2)$$

$$+ L\beta_t^{\max}(2G^2+\sigma^2)) + (\beta_t^{\max})^2 AG + L(\beta_t^{\max})^2\sigma^2 + 2L\beta_t^{\max}(\beta_t^{\max} + L(\beta_t^{\max})^2)(3\sigma^2 + 2G^2)]$$

$$\leq \hat{f}_\lambda^* - \hat{f}_\lambda(x_1) + (\beta_t^{\max})^2 \sum_{t=1}^{N}[(1 - \frac{L}{2}\beta_t^{\max})(L^2\beta_t^{\max}(G^2+\sigma^2) + L(2M_\rho^2+\sigma^2)) + AG + L\sigma^2$$

$$+ 2L(1 + L\beta_t^{\max})(3\sigma^2 + 2G^2)].$$

Rearrange it, we obtain

$$\mathbb{E}[\|\nabla \hat{f}_\lambda(x_z)\|_2^2] \leq \frac{\hat{f}_\lambda^* - \hat{f}_\lambda(x_1) + N(\beta_t^{\max})^2 C}{N(\beta_t^{\min} - \frac{L}{2}(\beta_t^{\max})^2)},$$

where

$$C = (1 - \frac{L}{2}\beta_t^{\max})(L^2\beta_t^{\max}(G^2+\sigma^2) + L(2G^2+\sigma^2)) + AG + L\sigma^2 + 2L(1 + L\beta_t^{\max})(3\sigma^2 + 2G^2).$$

$\square$

**Remark 2.** *In Theorem 3, we have proven the finite-sample analysis of Algorithm 1 with Option II, i.e., constant stepsizes and randomly picked solution. Note that the error bound in Eq. (12) can be simplified as $\mathcal{O}(1/(N\beta_t^{\min})) + \mathcal{O}(\beta_t^{\max})$. Especially, if $\beta_t^{\max} = \beta_t^{\min} = \beta_t^\theta = \beta_t^y$ are set to be $\Theta(1/\sqrt{N})$, then the convergence rate of Option II in Algorithm 1 is $\mathcal{O}(1/\sqrt{N})$.*

## B Proofs in Convergence Analysis of Nonconvex BSG

This section includes proof of Lemma 2 and Algorithm 2. We first provide the pseudo-code for nonconvex BSG method as Algorithm 2.

### B.1 Proof of Lemma 2

*Proof of Lemma 2.* We prove Lemma 2 for the case of discrete $\xi$, note that the proof still holds for the case of continuous $\xi$ just by using probability density function to replace probability distribution. Without the loss of generality, we assume a fixed update order in Algorithm 2: $\pi_t^i = i$, for all $i$ and $t$. Let $\Xi_t = \{\xi_{k,1}, \xi_{k,2}, \ldots, \xi_{k,m_t}\}$ be any mini-batch samples in the $t$-th iteration. Let $\tilde{g}_{\Xi_t,t}^i = \frac{1}{m_t}\sum_{j=1}^{m_t} \nabla_{x^i} F(x_{t+1}^{<i}, x_t^{\geq i}; \xi_{t,j})$ and $g_{\Xi_t,t}^i = \nabla_{x^i} f(x_{t+1}^{<i}, x_t^{\geq i})$, and $x_{\Xi_t,t+1}^i = x_t^i - \beta_t^i \tilde{g}_{\Xi_t,t}^i$. Then we have

$$\mathbb{E}[\tilde{g}_{\Xi_t,t}^i | \Xi_{[t-1]}] = \mathbb{E}_{\Xi_t}\left[\frac{1}{m_t}\sum_{j=1}^{m_t} \nabla_{x^i} F(x_{\Xi_t,t+1}^{\leq i}, x_t^{\geq i}; \xi_{t,j})\right]$$

$$= \sum_{\xi_1,\ldots,\xi_{m_t}} \Pr(\Xi_t = \{\xi_1, \xi_2, \ldots, \xi_{m_t}\})\frac{1}{m_t}\sum_{j=1}^{m_t} \nabla_{x^i} F(x_{\Xi_t,t+1}^{\leq i}, x_t^{\geq i}; \xi_j), \quad (26)$$

---
**Algorithm 2** The nonconvex BSG Algorithm
---
**Input:** Initial point $x_1 \in \mathbb{R}^n$, stepsizes $\{\beta_t^i : i = 1, \cdots, b\}_{k=1}^\infty$, positive integers $\{m_t\}_{k=1}^\infty$ that indicate the mini-batch sizes, and iteration limit $N$.

1: **for** $k = 1, 2, \ldots, N$ **do**
2:     Sample mini batch $\Xi_t = \{\xi_{t,1}, \xi_{t,2}, \ldots, \xi_{t,m_t}\}$.
3:     Specify update order $\pi_t^i = i$, $i = 1, \cdots, b$, or randomly shuffle $\{i = 1, \cdots, b\}$ to $\{\pi_t^1, \pi_t^2, \ldots, \pi_t^b\}$.
4:     **for** $i = 1, 2, \ldots, b$ **do**
5:         Compute the stochastic partial gradient for the $\pi_t^i$th block as

$$\tilde{g}_t^i = \frac{1}{m_t} \sum_{j=1}^{m_t} \nabla_{x^{\pi_t^i}} F(x_{t+1}^{\pi_t^{<i}}, x_t^{\pi_t^{\geq i}}; \xi_{t,j}).$$

6:         Update $\pi_t^i$th block

$$x_{t+1}^{\pi_t^i} = x_k^{\pi_t^i} - \beta_t^i \tilde{g}_t^i.$$

7:     **end for**
8: **end for**
9: Return $\bar{x}_N = x_z$ randomly according to

$$\Pr(z = t) = \frac{\beta_t^{\min} - \frac{L}{2}(\beta_t^{\max})^2}{\sum_{t=1}^N (\beta_t^{\min} - \frac{L}{2}(\beta_t^{\max})^2)}, \quad t = 1, \ldots, N. \tag{25}$$
---

and

$$\begin{aligned}
\mathbb{E}[g_{\Xi_t,t}^i | \Xi_{[t-1]}] &= \mathbb{E}_{\Xi_t} \left[ \nabla_{x^i} f(x_{\Xi_t,t+1}^{\leq i}, x_t^{\geq i}) \right] \\
&= \sum_{\xi_1', \ldots, \xi_{m_t}'} \Pr(\Xi_t' = \{\xi_1', \xi_2', \ldots, \xi_{m_t}'\}) \nabla_{x^i} f(x_{\Xi_t',t+1}^{\leq i}, x_t^{\geq i}) \\
&= \sum_{\xi_1', \ldots, \xi_{m_t}'} \Pr(\Xi_t' = \{\xi_1', \xi_2', \ldots, \xi_{m_t}'\}) \sum_{\xi_l} \Pr(\xi = \xi_l) \nabla_{x^i} F(x_{\Xi_t',t+1}^{\leq i}, x_t^{\geq i}; \xi_l) \\
&= \sum_{\xi_1', \ldots, \xi_{m_t}'} \Pr(\Xi_t' = \{\xi_1', \xi_2', \ldots, \xi_{m_t}'\}) \sum_{\xi_1, \ldots, \xi_{m_t}} \Pr(\Xi_t = \{\xi_1, \xi_2, \ldots, \xi_{m_t}\}) \\
&\quad \frac{1}{m_t} \sum_{j=1}^{m_t} \nabla_{x^i} F(x_{\Xi_t',t+1}^{\leq i}, x_t^{\geq i}; \xi_j). \tag{27}
\end{aligned}$$

Combine Eq. (26) and Eq. (27), we can obtain the expectation of $\Delta_t^i$ as

$$\begin{aligned}
\mathbb{E}[\Delta_t^i | \Xi_{[t-1]}] &= \mathbb{E}[\tilde{g}_t^i - g_t^i | \Xi_{[t-1]}] \\
&= \sum_{\xi_1', \ldots, \xi_{m_t}'} \Pr(\Xi_t' = \{\xi_1', \xi_2', \ldots, \xi_{m_t}'\}) \sum_{\xi_1, \ldots, \xi_{m_t}} \Pr(\Xi_t = \{\xi_1, \xi_2, \ldots, \xi_{m_t}\}) \\
&\quad \frac{1}{m_t} \sum_{j=1}^{m_t} (\nabla_{x^i} F(x_{\Xi_t,t+1}^{\leq i}, x_t^{\geq i}; \xi_j) - \nabla_{x^i} F(x_{\Xi_t',t+1}^{\leq i}, x_t^{\geq i}; \xi_j)) \\
&= \sum_{\xi_1', \ldots, \xi_{m_t}'} \sum_{\xi_1, \ldots, \xi_{m_t}} \Pr(\Xi_t' = \{\xi_1', \xi_2', \ldots, \xi_{m_t}'\}) \Pr(\Xi_t = \{\xi_1, \xi_2, \ldots, \xi_{m_t}\}) \\
&\quad \frac{1}{m_t} \sum_{j=1}^{m_t} (\nabla_{x^i} F(x_{\Xi_t,t+1}^{\leq i}, x_t^{\geq i}; \xi_j) - \nabla_{x^i} F(x_{\Xi_t',t+1}^{\leq i}, x_t^{\geq i}; \xi_j)). \tag{28}
\end{aligned}$$

Note that, since the objection function is Lipschitz smoothness, $F(x; \xi)$ is also Lipschitz smoothness if $\Pr(\xi) > 0$, and we use $L$ to denote the maximum Lipschitz constant for all $\Pr(\xi) > 0$. Similarly,

we can also obtain the gradient of $F(x;\xi)$ is also bounded using same analysis, and we use $G$ to denote the maximum bound for all $\Pr(\xi) > 0$. Using these two fact, we have

$$
\begin{aligned}
&\|\nabla_{x^i} F(x_{\Xi_t,t+1}^{\leq i}, x_t^{\geq i};\xi_j) - \nabla_{x^i} F(x_{\Xi_t',t+1}^{\leq i}, x_t^{\geq i};\xi_j)\|_2 \\
\leq& L \|x_{\Xi_t,t+1}^{\leq i} - x_{\Xi_t',t+1}^{\leq i}\|_2 \\
\leq& \sum_{l<i} L \|x_{\Xi_t,t+1}^l - x_{\Xi_t',t+1}^l\|_2 \\
\leq& \sum_{l<i} L \beta_t^l \|\tilde{g}_{\Xi_t,t}^l - \tilde{g}_{\Xi_t',t}^l\|_2 \\
\leq& 2LbG\beta_t^{\max}.
\end{aligned}
\tag{29}
$$

Combine Eq. (28) and Eq. (29), we complete the proof as

$$
\begin{aligned}
&\mathbb{E}[\Delta_t^i|\mathbf{\Xi}_{[t-1]}] \\
\leq& \sum_{\xi_1',\ldots,\xi_{m_t}'} \sum_{\xi_1,\ldots,\xi_{m_t}} \Pr(\Xi_t' = \{\xi_1',\xi_2',\ldots,\xi_{m_t}'\}) \Pr(\Xi_t = \{\xi_1,\xi_2,\ldots,\xi_{m_t}\}) 2LbG\beta_t^{\max} \\
=& 2LbG\beta_t^{\max},
\end{aligned}
$$

where the last equation follows from the Law of total probability. This completes the proof. $\qquad\square$

### B.2 Proof of Theorem 2

Let $\beta_t^{\max} := \max_i \beta_t^i$, $\beta_t^{\min} := \min_i \beta_t^i$. To establish the convergence rate analysis, we start with Lemma 5.

**Lemma 5.** *Let $u_k$ be a random vector that only depends on $\mathbf{\Xi}_{[t-1]}$. If $u_t$ is independent of $\Delta_t^i$, then*

$$
\mathbb{E}[\langle u_t, \Delta_t^i \rangle] \leq A\beta_t^{\max} \mathbb{E}[\|u_t\|_2],
$$

*where $A$ is defined in Eq (9).*

Now, it is ready to discuss the main convergence properties of the nonconvex Cyclic SBCD algorithm (Algorithm 2) and provide the rate of convergence for that.

*Proof of Lemma 5.* We can obtain the result of Lemma 5 by follows

$$
\begin{aligned}
\mathbb{E}[\langle u_t, \Delta_t^i \rangle] =& \mathbb{E}_{\mathbf{\Xi}_{[t-1]}} \left[ \mathbb{E}[\langle u_t, \Delta_t^i \rangle | \mathbf{\Xi}_{[t-1]}] \right] \\
\overset{(a)}{=}& \mathbb{E}_{\mathbf{\Xi}_{[t-1]}} \left[ \langle \mathbb{E}[u_t | \mathbf{\Xi}_{[t-1]}], \mathbb{E}[\Delta_t^i | \mathbf{\Xi}_{[t-1]}] \rangle \right] \\
\leq& \mathbb{E}_{\mathbf{\Xi}_{[t-1]}} \left[ \|\mathbb{E}[u_t | \mathbf{\Xi}_{[t-1]}]\|_2 \cdot \|\mathbb{E}[\Delta_t^i | \mathbf{\Xi}_{[t-1]}]\|_2 \right] \\
\leq& A\beta_t^{\max} \mathbb{E}_{\mathbf{\Xi}_{[t-1]}} \left[ \|\mathbb{E}[u_t | \mathbf{\Xi}_{[t-1]}]\|_2 \right] \\
\overset{(b)}{\leq}& A\beta_t^{\max} \mathbb{E}[\|u_t\|_2],
\end{aligned}
$$

where (a) follows from the conditional independence between $u_t$ and $\Delta_t^i$, and (b) follows from Jensen's inequality. $\qquad\square$

*Proof of Theorem 2.* From the Lipschitz smoothness, it holds that

$$
\begin{aligned}
&f(x_{t+1}^{\leq i}, x_t^{>i}) - f(x_{t+1}^{<i}, x_t^{\geq i}) \\
\leq& \langle g_t^i, x_{t+1}^i - x_t^i \rangle + \frac{L}{2} \|x_{t+1}^i - x_t^i\|_2^2 \\
=& -\beta_t^i \langle g_t^i, \tilde{g}_t^i \rangle + \frac{L}{2}(\beta_t^i)^2 \|\tilde{g}_t^i\|_2^2 \\
=& -(\beta_t^i - \frac{L}{2}(\beta_t^i)^2)\|g_t^i\|_2^2 + \frac{L}{2}(\beta_t^i)^2 \|\Delta_t^i\|_2^2 - (\beta_t^i - L(\beta_t^i)^2)\langle g_t^i, \Delta_t^i \rangle
\end{aligned}
\tag{30}
$$

where all the equations follow the definition of $\Delta_t^i$ and the update law of Algorithm 2.

Summing Eq. (30) over $i$, then we obtain

$$f(x_{t+1}) - f(x_t) \leq -\sum_{i=1}^{b}(\beta_t^i - \frac{L}{2}(\beta_t^i)^2)\|g_t^i\|_2^2 + \sum_{i=1}^{b}\frac{L}{2}(\beta_t^i)^2\|\Delta_t^i\|_2^2 - \sum_{i=1}^{b}(\beta_t^i - L(\beta_t^i)^2)\langle g_t^i, \Delta_t^i\rangle. \tag{31}$$

Use Lemma 5, we also have the following fact,

$$\mathbb{E}[\langle g_t^i, \Delta_t^i\rangle] \leq \beta_t^{\max} AG. \tag{32}$$

Taking expectation over Eq. (31), we have

$$\mathbb{E}[f(x_{t+1})] - \mathbb{E}[f(x_t)]$$
$$\leq -\sum_{i=1}^{b}(\beta_t^i - \frac{L}{2}(\beta_t^i)^2)\mathbb{E}[\|g_t^i\|_2^2] + \sum_{i=1}^{b}(\beta_t^i - L(\beta_t^i)^2)\beta_t^{\max} AG + \sum_{i=1}^{b}\frac{L}{2}(\beta_t^i)^2\mathbb{E}[\|\Delta_t^i\|_2^2]$$
$$\leq -(\beta_t^{\min} - \frac{L}{2}(\beta_t^{\max})^2)\sum_{i=1}^{b}\mathbb{E}[\|g_t^i\|_2^2] + \sum_{i=1}^{b}\left((\beta_t^{\max})^2 AG + \frac{L}{2}(\beta_t^i)^2\sigma^2\right), \tag{33}$$

where the first inequality follows from Eq. (32), and the second inequality follows from the boundedness of $\mathbb{E}[\|\Delta_t^i\|_2]$ and $\mathbb{E}[\|g_t^i\|_2]$.

Rearranging Eq. (33), we obtain

$$(\beta_t^{\min} - \frac{L}{2}(\beta_t^{\max})^2)\sum_{i=1}^{b}\mathbb{E}[\|g_t^i\|_2^2] \leq \mathbb{E}[f(x_t)] - \mathbb{E}[f(x_{t+1})] + \sum_{i=1}^{b}\left((\beta_t^{\max})^2 AG + \frac{L}{2}(\beta_t^i)^2\sigma^2\right). \tag{34}$$

Also, we have

$$\mathbb{E}[\|\nabla_{x^i} f(x_t)\|_2] \leq \mathbb{E}[\|\nabla_{x^i} f(x_t) - g_t^i\|_2] + \mathbb{E}[\|g_t^i\|_2]$$
$$\overset{(a)}{\leq} L\mathbb{E}[\|x_{t+1}^{<i} - x_t^{<i}\|_2] + \mathbb{E}[\|g_t^i\|_2]$$
$$\overset{(b)}{\leq} L\mathbb{E}\left[\sqrt{\sum_{j<i}\|\beta_t^j \tilde{g}_t^j\|_2^2}\right] + \mathbb{E}[\|g_t^i\|_2]$$
$$\leq L\beta_t^{\max}\mathbb{E}\left[\sqrt{\sum_{j<i}\|\tilde{g}_t^j\|_2^2}\right] + \mathbb{E}[\|g_t^i\|_2]$$
$$\overset{(c)}{\leq} L\beta_t^{\max}\sqrt{\mathbb{E}[\sum_{j<i}\|\tilde{g}_t^j\|_2^2]} + \mathbb{E}[\|g_t^i\|_2]$$
$$\overset{(d)}{\leq} L\beta_t^{\max}\sqrt{\sum_{j<i}(G^2 + \sigma^2)} + \mathbb{E}[\|g_t^i\|_2], \tag{35}$$

where (a) follows from the Lipschitz smoothness of $f$, (b) follows from $x_{t+1}^j = x_t^j - \beta_t^j \tilde{g}_t^j$, (c) follows from Jenson's inequality, and (d) follows from the boundedness of gradient and boundedness of variance.

Summing Eq. (35) over $i$, we can obtain

$$\mathbb{E}[\|\nabla f(x_t)\|_2^2] = \sum_{i=1}^{b}\mathbb{E}[\|\nabla_{x^i} f(x_t)\|_2] \leq \sum_{i=1}^{b} L\beta_t^{\max}\sqrt{\sum_{j<i}(G^2 + \sigma^2)} + \sum_{i=1}^{b}\mathbb{E}[\|g_t^i\|_2]. \tag{36}$$

Combine Eq. (36) with Eq. (34), we can obtain

$$(\beta_t^{\min} - \frac{L}{2}(\beta_t^{\max})^2)\mathbb{E}[\|\nabla f(x_t)\|_2^2]$$

$$\leq (\beta_t^{\min} - \frac{L}{2}(\beta_t^{\max})^2) \sum_{i=1}^b L\beta_t^{\max}\sqrt{\sum_{j<i}(G^2+\sigma^2)} + (\beta_t^{\min} - \frac{L}{2}(\beta_t^{\max})^2)\sum_{i=1}^b \mathbb{E}[\|g_t^i\|_2]$$

$$\leq \mathbb{E}[f(x_t)] - \mathbb{E}[f(x_{t+1})] + (\beta_t^{\min} - \frac{L}{2}(\beta_t^{\max})^2)\sum_{i=1}^b L\beta_t^{\max}\sqrt{\sum_{j<i}(G^2+\sigma^2)}$$

$$+ \sum_{i=1}^b \left((\beta_t^{\max})^2 AG + \frac{L}{2}(\beta_t^i)^2\sigma^2\right), \tag{37}$$

where the first inequality follows from substituting Eq. (36) into the left-hand side of Eq. (34), and the second inequality follows from substituting Eq. (36) into the right-hand side of Eq. (34).

Summing Eq. (37) over $t$, we have

$$\sum_{t=1}^N (\beta_t^{\min} - \frac{L}{2}(\beta_t^{\max})^2)\mathbb{E}[\|\nabla f(x_t)\|_2^2]$$

$$\leq f(x_1) - f(x^*)$$

$$+ \sum_{t=1}^N \left[(\beta_t^{\min} - \frac{L}{2}(\beta_t^{\max})^2)\sum_{i=1}^b L\beta_t^{\max}\sqrt{\sum_{j<i}(G^2+\sigma^2)} + \sum_{i=1}^b \left((\beta_t^{\max})^2 AG + \frac{L}{2}(\beta_t^i)^2\sigma^2\right)\right]$$

$$\leq f(x_1) - f(x^*) + \sum_{t=1}^N (\beta_t^{\max})^2 C_t. \tag{38}$$

where $C_t$ is

$$C_t = (1 - \frac{L}{2}\beta_t^{\max})\sum_{i=1}^b L\sqrt{\sum_{j<i}(G^2+\sigma^2)} + b\left(AG + \frac{L}{2}\sigma^2\right).$$

Using the probability distribution of $R$ given in Eq. (25), we completes the proof. $\qquad\square$

*Proof of Corollary 1.* Combine these conditions with Eq. (38), we have

$$\sum_{t=1}^N (\beta^{\min} - \frac{L}{2}(\beta^{\max})^2)\mathbb{E}[\|\nabla f(x_t)\|_2^2] \leq f(x_1) - f(x^*) + N(\beta^{\max})^2 C,$$

where $C$ is

$$C = (1 - \frac{L}{2}\beta^{\max})\sum_{i=1}^b L\sqrt{\sum_{j<i}(G_j^2+\sigma^2)} + b\left(AG + \frac{L}{2}\sigma^2\right).$$

Using the probability distribution of $z$ given in Eq. (25), we can obtain

$$\mathbb{E}\left[\|\nabla f(x_z)\|_2^2\right] \leq \frac{f(x_1) - f^* + N(\beta^{\max})^2 C}{N(\beta^{\min} - \frac{L}{2}(\beta^{\max})^2)}.$$

Thus, we can reach the rate of convergence of $\mathcal{O}(1/\sqrt{N})$ by setting $\beta^{\min} = \beta^{\max} = \mathcal{O}(1/\sqrt{N})$. $\quad\square$

---
**Algorithm 3** Risk-Sensitive Randomized Coordinate Descent Policy Gradient (RCPG)
---
1: **Input:** Stepsizes $\{\beta_t^\theta\}$ and $\{\beta_t^y\}$, let $\beta_t^{\max} = \max\{\beta_t^\theta, \beta_t^y\}$.
   **Option I:** $\{\beta_t^\theta\}$ and $\{\beta_t^y\}$ satisfy the Robbins-Monro condition.
   **Option II:** $\beta_t^\theta$ and $\beta_t^y$ are set to be constants.
2: **for** episode $t = 1, \ldots, N$ **do**
3:    **for** time step $k = 1, \ldots, \tau_t$ **do**
4:       Compute $a_k \sim \pi_\theta(a|s_k)$, observe $r_k, s_{k+1}$.
5:    **end for**
6:    Compute

$$R_t = \sum_{k=1}^{\tau_t} r_k$$

$$\omega_t(\theta_t) = \sum_{k=1}^{\tau_t} \nabla_\theta \ln \pi_{\theta_t}(a_k|s_k).$$

7:    Randomly select $i_t \in \{1, 2\}$ with distribution $[0.5, 0.5]$. If $i_t = 1$,

$$y_{t+1} = y_t + \beta_t \left( 2R_t + \frac{1}{\lambda} - 2y_t \right),$$

$$\theta_{t+1} = \theta_t.$$

     else

$$y_{t+1} = y_t,$$

$$\theta_{t+1} = \theta_t + \beta_t \left( 2y_t R_t - (R_t)^2 \right) \omega_t(\theta_t).$$

8: **end for**
9: **Output** $\bar{x}_N$:
   **Option I:** Set $\bar{x}_N = x_N$.
   **Option II:** Set $\bar{x}_N = x_z$, where $z$ is uniformly drawn from $\{1, 2, \ldots, N\}$.
---

## C  RCPG and SGA Algorithm

### C.1  Randomized Stochastic Block Coordinate Descent Algorithm

We propose the randomized stochastic block coordinate descent algorithm as Algorithm 3. Note that we also use the same notation about gradient from Eq. (6) and Eq. (7) with very a tiny difference in practical, where $y_{t+1} = y_t$ in Eq. (7).

Note that, the main difference between Cyclic SBCD and Randomized SBCD is that: at each iteration, Cyclic SBCD cyclically updates all blocks of variables, and the later updated blocks depending on the early updated blocks; while Randomized SBCD randomly chooses one block of variables to update.

### C.2  Risk-Sensitive Stochastic Gradient Ascent Policy Gradient

We also proposed risk-sensitive stochastic gradient Ascent policy gradient as Algorithm 4.

## D  Details of the Experiments

The parameter settings for portfolio management domain are as follows: $\tau = 50$, $r_l = 1.001$, $r_{nl}^{high} = 2$, $r_{nl}^{low} = 1.1$, $p_{risk} = 0.05$, $p_{switch} = 0.1$, $W = 4$, $\eta = 0.2$, startup cash $\$100,000$.

The parameter settings of American-style Option domain are as follows: $K_{put} = 1$, $K_{call} = 1.5$, $x_0 = 1.25$, $f_u = 9/8$, $f_d = 8/9$, $p = 0.45$, $\tau = 20$.

---

**Algorithm 4** Risk-Sensitive Stochastic Gradient Ascent Policy Gradient (SGA)

---

1: **Input:** Stepsizes $\{\beta_t^\theta\}$ and $\{\beta_t^y\}$, let $\beta_t^{\max} = \max\{\beta_t^\theta, \beta_t^y\}$.
   **Option I:** $\{\beta_t^\theta\}$ and $\{\beta_t^y\}$ satisfy the Robbins-Monro condition.
   **Option II:** $\beta_t^\theta$ and $\beta_t^y$ are set to be constants.
2: **for** episode $t = 1, \dots, N$ **do**
3:   **for** time step $k = 1, \dots, \tau_t$ **do**
4:      Compute $a_k \sim \pi_\theta(a|s_k)$, observe $r_k, s_{k+1}$.
5:   **end for**
6:   Compute

$$R_t = \sum_{k=1}^{\tau_t} r_k$$

$$\omega_t(\theta_t) = \sum_{k=1}^{\tau_t} \nabla_\theta \ln \pi_{\theta_t}(a_k|s_k).$$

7:   Update parameters,

$$y_{t+1} = y_t + \beta_t \left( 2R_t + \frac{1}{\lambda} - 2y_t \right),$$

$$\theta_{t+1} = \theta_t + \beta_t \left( 2y_t R_t - (R_t)^2 \right) \omega_t(\theta_t).$$

8: **end for**
9: **Output** $\bar{x}_N$:
   **Option I:** Set $\bar{x}_N = x_N$.
   **Option II:** Set $\bar{x}_N = x_z$, where $z$ is uniformly drawn from $\{1, 2, \dots, N\}$.

---

The parameter settings of optimal stopping domain are as follows: $x_0 = 1.25$, $f_u = 2$, $f_d = 0.5$, $p = 0.65$, $\tau = 20$.