[Reviews · NeurIPS 2018]

Reviewer 1



This work is deals with risk measure in RL/Planning, specifically, risk measures that are based on trajectory variance. Unlike the straight forward approach that is taken in previous works, such as Tamar et al. 2012, 2014 or 2; Prashanth and Ghavamzadeh, 2013, in this work multiple time scale stochastic approximation (MTSSA) is not used. The authors argue that MTSSA is hard to tune and has a slow convergence rate. Instead, the authors propose an approach which is use Block Coordinate Descent. This approach is based on coordinate descent where during the optimization process, not all the coordinates of the policy parameter are optimized, but only a subset of them. After assumption relatively standard assumptions (ergodicity of the MDP and boundedness of rewards), the authoers provide a convergence proof (to a local optima). Afterward, the authors provide a finite sample analysis of of Nonconvex BSG Algorithms, where their provided RL algorithm belongs to this family algorithms. I followed the mathematics and it seems correct (the sketch). Finally, the authors provide experiments, in the lines of Tamar et al. 2012 and 2014 and show their algorithm validity. In general I like this work. It provides additional methods for optimizing the mean-variance trade-off. Also, the finite sample analysis in this case, where one do not need the finite sample analysis of multiple time scale stochastic approximation is a good technique in the general case of BCD. My only concern is that I'm not sure about the argument of the authors that such method provide a faster convergence with respect to Tamar et al. 2012 for example (or that Tamar et al. convergence rate is low). I would like to have clarification regarding this. I refer also the authors to a recent paper "Concentration Bounds for Two Timescale Stochastic Approximation with Applications to Reinforcement Learning" of Dalal et al. that show how to compute finite sample analysis in the case of multiple time scales problems. ---- After author feedback -------- I read the feedback. I thank the authors for it. I maintain the same feedback and score.

Reviewer 2



The paper proposes a nice algorithm for risk-aware reinforcement learning. The original objective function is reformulated through Fenchel dual. This formulation gets rid of the two-sample issue in the original formulation. Finite sample analysis is also proposed. I think it is a nice work. In RL, the independent sampling issue arises in different settings. For example, in policy evaluation, when using MSPBE as an objective, the gradient method would also derive sth like the product of two expectation terms. The most common strategy so far is to use two time scale updating, where usually two updating rules need to be updated at different speeds and could be very sensitive. This work is an important first step, though the Block coordinate method still requires two step-sizes. I think it should be easier to tune (i.e. maybe same step sizes for both). From theoretical perspective, at least the work is able to provide a finite sample analysis, which is not common in the big RL area. And, this work provides a new perspective to resolve the two time scale problem, though it is in risk-aware RL, it could be a good reference for other settings. However, I still think careful designed experiments in classical RL domains can be added to make more people feel interesting. Particularly, for a new algorithm, it is better to use simpler domain to let first-time-readers know what the domain's difficulties are and make it clear about how well the algorithm work with a thorough parameter sweep.

Reviewer 3



Summary: This paper studies risk-sensitive sequential decision-making, starting from the mean-variance function. The problem with standard stochastic gradient descent on this objective is that it needs multi-time-scale stochastic approximations, for which the learning rates are hard to tune, and which lack finite-sample guarantees. The authors propose a new policy search algorithm based on a (Fenchel) dual formulation of the mean-variance objective. In particular, they provide finite-sample analysis of block coordinate descent on a non-convex objective (which the authors claim has never been reported before), and apply this to the policy gradient setting. Empirical results on three (small) domains show similar mean return with smaller variance compared to competing (risk-sensitive) methods. Strength: see conclusion. Comments: - Results (5.2 and 5.3). I think you are a bit optimistic in your results interpretation. For example, in line 280 (about Fig 1b), your approach does achieve smaller variance, but I don’t think you really achieve higher mean return. Then, in line 288 (about Fig 1c) you indicate that your method is ‘not affecting the return’. I think the difference in means in Fig 1c (where your method performs worse) is actually larger than in Fig 1b (where your method performs better), so I don’t think you can say for 1b that you outperform, while saying in 1c you don’t do worse. I think you do almost equal in both, and for other claims you would need a statistical hypothesis test. - Figure 1: Why are the means not in the center of the Gaussians you draw? Moreover, I think the captioning lacks some information (such as the abbreviations). These are aggregate distributions from the entire learning process right, i.e., all returns sampled? - The graphs look nice, but the sigma’s are a bit hard to compare, especially when the means differ. I would prefer some numerical values of means and standard deviations in a table, if you have space. - line 255: Out of curiosity: Could your algorithm be extended to the actor-critic class as well? Would your algorithm benefit from variance reduction techniques (for the policy gradient), such as learning a value function baseline? Conclusion: My background is in planning and reinforcement learning/sequential decision making under uncertainty, but not in theoretical analysis of non-convex optimization. I therefore cannot fully judge the importance and technical correctness of the contribution in Section 4 (although it appears to be sound). I will mostly judge the RL contribution. Risk-sensitive is a longer research line in RL, which I think will get more attention after the success of the distributional RL paper [1] in high-dimensional problems as well. II do see a contribution in this paper to the risk-sensitive RL literature, in the reformulation (Fenchel dual) of the mean-variance objective. The results look better than competing risk-sensitive approaches, and the paper is well-written and motivated as well. The RL part of this paper is therefore moderately novel, but conceptually sound and with promising results. After rebuttal: The authors adequately addressed my concerns. I believe it is important to move Table 1 to the main paper, and plot the means in Figure 1 (or else explain what the current markers mean, they clearly confuse me). [1] Bellemare, Marc G., Will Dabney, and Rémi Munos. "A Distributional Perspective on Reinforcement Learning." International Conference on Machine Learning. 2017.

Reviewer 4



The paper "A Block Coordinate Ascent Algorithm for Mean-Variance Optimization" proposes a new algorithm for reinforcement learning with a mean-variance objective that has a hard constraint on the reward variance and analyses the convergence of the algorithm in both infinite and finite horizon cases. The derivations for the new algorithm use a transformation based on a Fenchel dual (to transform a quadratic term into a linear term plus terms with an added additional variable) enabling gradient descent to be applied on the transformed Lagrangian dual of the original objective. The contribution w.r.t. the proofs and the new algorithm appear solid. The paper is in general well written. There are some typos that should be fixed. The main problem in the paper is the reporting of the experimental results. The representation used for the policy is not discussed, the variance limit is not defined, and it is unclear whether the reported results are statistically significant. The main claims about related work are as follows: "1) Most of the analyses of ODE-based methods are asymptotic, with no sample complexity analysis." Most are asymptotic? The paper needs to discuss those cases where the analysis is not asymptotic. This is discussed later in the paper? Should be already mentioned here. "2) It is well-known that multi-time-scale approaches are sensitive to the choice of the stepsize schedules, which is a non-trivial burden in real-world problems." "3) The ODE approach does not allow extra penalty functions. Adding penalty functions can often strengthen the robustness of the algorithm, encourages sparsity and incorporates prior knowledge into the problem [Hastie et al., 2001]." Claims 1) - 3) seem correct and the proposed algorithm could offer some improvement over these previous approaches. Algorithm 1: Each episode starts from the initial state s_1 until ending in the recurrent state s^*. It could help the reader to give an explanation why this is done so? Is this absolutely necessary for the analysis of the algorithm? EXPERIMENTS: In the experiments, it is unclear what kind of representation the policy follows. It cannot be tabular since some of the problems have continuous states. It is also unclear whether the form of the policy influences the performance of the algorithms. \xi, the limit on the expected variance of the rewards (Equation 1), should be defined for each problem. The statistical significance of the results is now described only in textual form. For example, "The results indicate that MVP yields a higher mean return with less variance compared to the competing algorithms." Firstly, according to my understanding the objective is not to minimize variance but to limit it, at least according to Equation 1. As such the variance of the comparison methods does not matter if it remains under the limit (which is not defined). Of course the variance is of separate interest for understanding how far each algorithm is from the limit. I recommend reporting whether the variance limit is exceeded and by how much. Furthermore, I recommend performing statistical significance testing on both the mean and variance. LANGUAGE: In some places there is some unclarity that should be fixed. For example, in "yet almost all the reported analysis of these algorithms are asymptotic [Xu and Yin, 2015]." almost all? Which reported analysis is not asymptotic? "In risk-sensitive mean-variance optimization MDPs, the objective is often to maximize J(θ) with a variance constraint, i.e.," *often* ? Earlier in the paper there is a discussion on related work but it was not made clear what exactly the objective in those papers is? Smaller typos: Line 85: "the third time-scale is to optimizes over" -> the third time-scale optimizes over Line 48: "2 Backgrounds" -> "2 Background" ? Line 116: "and derives novel algorithms" -> and derive novel algorithms Line 188: "for which there is no established results" -> for which there are no established results Line 203: "Without the loss of generality" should be usually "Without loss of generality" Line 205: "as as" -> as Line 206: "approximate error" -> approximation error Line 219: "Now it is ready to provide" -> Now we provide